# Nanoparticle Deposition in Rhythmically Moving Acinar Models with Interalveolar Septal Apertures

**DOI:** 10.3390/nano9081126

**Published:** 2019-08-04

**Authors:** Jinxiang Xi, Mohamed Talaat

**Affiliations:** 1Department of Biomedical Engineering, University of Massachusetts, Lowell, MA 01854, USA; 2Department of Biomedical Engineering, California Baptist University, Riverside, CA 92504, USA; 3Department of Aerospace, Industrial, and Mechanical Engineering, California Baptist University, Riverside, CA 92504, USA

**Keywords:** nanoparticle alveolar deposition, nanomedicine, rhythmic wall motion, interalveolar septal wall, pore of Kohn, collateral ventilation

## Abstract

Pulmonary delivery of nanomedicines has been extensively studied in recent years because of their enhanced biocompatibility, sustained-release properties, and surface modification capability. The lung as a target also offers many advantages over other routers, such as large surface area, noninvasive, quick therapeutic onset, and avoiding first-pass metabolism. However, nanoparticles smaller than 0.26 µm typically escape phagocytosis and remain in the alveoli for a long time, leading to particle accumulation and invoking tissue responses. It is imperative to understand the behavior and fates of inhaled nanoparticles in the alveoli to reliably assess therapeutic outcomes of nanomedicines or health risk of environmental toxins. The objective of this study is to numerically investigate nanoparticle deposition in a duct-alveolar model with varying sizes of inter-alveolar septal apertures (pores). A discrete phase Lagrangian model was implemented to track nanoparticle trajectories under the influence of rhythmic wall expansion and contraction. Both temporal and spatial dosimetry in the alveoli were computed. Wall motions are essential for nanoparticles to penetrate the acinar region and deposit in the alveoli. The level of aerosol irreversibility (i.e., mixing of inhaled nanoparticles with residual air in the alveolar airspace) is determined by the particle diffusivity, which in turn, dictates the fraction of particles being exhaled out. When deposition in the upper airways was not considered, high alveolar deposition rates (74–95%) were predicted for all nanoparticles considered (1–1000 nm), which were released into the alveoli at the beginning of the inhalation. The pore size notably affects the deposition pattern of inhaled nanoparticles but exerts a low impact upon the total deposition fractions. This finding indicates that consistent pulmonary doses of nanomedicine are possible in emphysema patients if breathing maneuver with the same tidal volume can be performed.

## 1. Introduction

The integration of nanoscience and inhalation drug delivery offers exciting potentials to enhance the targeting, release, diagnostic, and therapeutic outcomes of drugs [1,2,3]. Various nanomedicines have been devised in recent years for local and systemic treatments of diseases, which include small molecules, macromolecules, peptide, protein, and genes. For instance, local applications of small molecules were tested to treat respiratory diseases, such as chronic obstructive pulmonary disease (COPD) and asthma [4,5]. Protein/peptide delivery to the lung is promising for both local therapy of respiratory diseases and systemic management of thrombosis or diabetes [6]. Targeted pulmonary delivery of genes to the disease site offers great potential for the treatment of genetic lung disorders, such as alpha-1-antitrypsin deficiency [7,8], cystic fibrosis [9], and asthma [10].

The human respiratory tract can be divided into several sections: Oropharyngeal region, tracheobronchial region, bronchioles, respiratory bronchioles, and alveoli. Each section has different morphology, dimension, type of epithelium cell, as well as different mechanisms for drug deposition, absorption, and clearance [11]. In the acinar region (alveoli), only particles smaller than 2 µm can get deposited, and only nanoparticles coated with albumin or phospholipids can translocate across the alveolus–capillary barrier, both chemicals naturally present in the alveolar epithelial lining fluid [12,13]. Particle clearance from the alveoli is predominantly mediated by macrophages, which only take up particles of 0.5–3 μm [14]. Particles 260 nm or smaller eludes the phagocytosis and cannot be easily cleared from the pulmonary region [15]. Such nanocarriers can remain in the lungs for up to 700 days [16], which either remain in the alveolar cells or translocate into the interstitium, causing long-term accumulations and invoking toxicity issues. It is therefore critical to understand the behavior and fate of inhaled nanoparticles inside the alveoli, to reliably establish the dose-outcome relationship for either inhaled therapeutic pharmaceuticals or environmental toxins.

Alveolar sacs are the smallest respiration units at the end of the respiratory tree. The alveoli exhibit a highly intricated architecture braced by interalveolar septal walls [17]. Circular or oval apertures (pores of Kohn) exist in the septum, allowing communications of airflow and liquids between neighboring alveoli [18]. The collateral ventilation equalizes pressures among alveoli and acts as an important mechanism in promoting alveolar recruitment [19] and preventing atelectasis (i.e., lung collapse) [20]. In lungs with emphysema, the pores of Kohn increase in both size and number [21,22,23]. As the emphysema deteriorates, the septal walls break down, leading to enlarged sac airspace and weakened lung elastin [24]. Aging can also cause the pores to enlarge and proliferate [25,26]. Knowledge of the influences from the interalveolar septal walls and pore size on acinar airflow and nanoparticle behaviors is of fundamental importance in accurately predicting pulmonary dosimetry, developing dose-response relationships, and devising more effective treatment strategies.

Theoretical and numerical studies have investigated airflow and particle deposition in the alveoli. Theoretical analysis of acinar deposition assumed particle diffusion and sedimentation in steady tubular flows, while neglected some critical factors, such as tidal breathing, geometry details, dynamic walls, and tidal breathing [27,28,29]. Using numerical methods, Kojic and Tsuda [30] elegantly demonstrated that particle deposition could be significantly underestimated using steady flow solutions to approximate tidal breathing. Kim et al. [31,32] demonstrated that the rhythmic wall motions are essential to match in vivo alveolar deposition data with predictions from the single-path-transport model. Kumar et al. [33] numerically investigated respiratory flows in honeycomb-like alveolar models and found recirculating stream traces in the alveolar airspace. Talaat and Xi [34] simulated particle dynamics in a terminal alveolus with rhythmic wall motions and reported significantly different particle motions from those in respiratory bronchioles or alveolar ducts [35,36,37,38]. In a terminal alveolus, the oscillating walls cause a particle to move forth and back, forming a multi-folding trajectory [34]. By contrast, in a respiratory bronchiole or alveolar duct, particles remain in the alveolar airspace for several cycles, rotating counterclockwise during inhalation and clockwise during exhalation [35,38]. Recent attempts to develop empirical correlations were also reported in alveolar models [34,38,39,40,41,42]. However, most studies have excluded the interalveolar septal walls and apertures (pores) for geometrical simplicity. In addition, reports of nanoparticle transport and deposition in alveolar sacs with moving walls are scarce, possibly due to the numerical challenges in fluid-wall-particle interactions, which include the dynamic mesh, random Brownian motion, multiscale velocities, and multi-physics of forces resulting from and acting on walls, fluid, and particles.

The objective of this study is to investigate the nanoparticle transport and deposition in a simplified alveolar model that comprises one cylindrical duct and four interconnecting spherical alveoli. Specific aims include: (1) To numerically simulate airflows and nanoparticle motions in alveolar models for different particle sizes (diffusivity), (2) to study the influences of the pore size on airflows and nanoparticle motions, (3) to quantify the alveolar deposition both temporally and spatially for different nanoparticle sizes, (4) to evaluate the effect of the pore size on alveolar deposition for nanoparticles representative of inhalation nanomedicine.

## 2. Methods

### 2.1. Alveolar Model and Wall Kinematics

Human lungs consist of 480 million or so alveoli. There are more than 10,000 alveoli in one acinar unit [17]. Even developing a complete model for a single acinar unit can be prohibitive. In this study, a four-alveoli model that was previously developed in Xi et al. [41] was adopted for its well-defined shape and dimension. Figure 1a shows the surface geometry of the four-alveoli model in transparent and solid views, while Figure 1b shows the cut-open view of the hollow model, with interalveolar septal apertures (pores) connecting adjacent alveoli. To help differentiate the four alveoli, the upper one was referred to as Sac 1, the left and right as Sac 2 and Sac 3, and the lower as Sac 4 (Figure 1a).

The rhythmic expansion and contraction of the alveolar wall were assumed to follow the chest [43,44], which moves a longer distance in the head-foot (*x*) and back-front (*y*) directions than the arm-arm (*z*) direction (i.e., *x:y:z* = 1:1:0.375). A constant ratio of the tidal volume (V_T_) to the functional residual capacity (FRC) of 23.3% was used (i.e., V_T_/FRC = 0.233) [45]. In-house code was written to specify the rhythmic wall expansion and contraction (Figure 1b). More details of the alveolar wall kinematics can be found in Talaat and Xi [34].

To evaluate the impact of the interalveolar septal aperture (pore) size on the alveolar deposition, seven geometrically similar models were developed with different pore sizes, i.e., 20 µm, 40 µm, 50 µm, 60 µm, 70 µm, 80 µm, and 90 µm. Figure 1c,d, and 1e display the cut view of the alveolar model with a pore size of 20 µm, 40 µm, and 80 µm, respectively. One model with complete septal destruction (no septal wall) was also developed for comparison purposes (Figure 1f).

### 2.2. Airflow and Nanoparticle Transport Models

Incompressible and isothermal airflow was assumed in this study. Based on the airflow speed of 0.3 mm/s and a characteristic length of 0.2 mm, the Reynolds number is around 0.004, indicating a laminar flow regime. Particles ranging from 1 nm to 1000 nm in diameter were investigated. For each case, multiple breathing cycles (5–6) were simulated, with the first cycle to establish the transient flow field in the alveolar airspace. To simulate the inhalation of a bolus of pharmaceutical particles, a group of 10,000 particles was released into the duct at 0.20 s at the second cycle. These particles were tracked until all deposited or exited the alveolar model. Based on the particle size ranging from 1 to 1000 nm, the Peclet number (Pe), which is the ratio of convection to diffusion, ranges from 0.01 to 2618. For 10-nm and 200-nm particles, Pe equals 1.1 and 262, respectively. Special attention was paid in the behaviors and fates of these two aerosols, as the first reacts equivalently to convection and diffusion, while the second represents the typical size of inhalation nanomedicine.

A discrete-phase Lagrangian tracking model was applied to follow the particle paths [46,47]. This model, enhanced with the near-wall treatment algorithm [48], has been demonstrated in our previous studies to agree with experimental deposition results in the extratropic airway for both nanoparticles [49] and micron aerosols [50,51]. The transport governing equations can be expressed as:(1)$$\frac{dv_{i}}{dt}\;=\;\frac{f}{\tau_{p}}\;\left(u_{i}-v_{i}\right)+g_{i}(1-\alpha )+f_{i,\;lift}+f_{i,Brownian}\;\mathrm{and}\;\frac{dx_{i}}{dt}=v_{i}(t)$$

Here, *v_i_* and *u_i_* are the local velocity of the particle and fluid, *g_i_* is the gravity, and *τ_p_* is the particle relaxation time expressed as *τ_p_* = C_c_*ρ_p_ d_p_*^2^/18*µ*, where *C_c_* is the Cunningham correction factor for nanoparticles [52]. The drag factor *f* is computed from the expression of Morsi and Alexander [53]. The effect of Brownian motion on nanoparticle trajectories is considered as an additional force per unit mass term at each time-step:(2)$$f_{i,Brownian}=\frac{\varsigma_{i}}{m_{d}}\sqrt{\frac{1}{{\tilde{D}}_{p}}\frac{2k^{2}T^{2}}{\Delta t}}$$
where $\varsigma_{i}$ is a random number generated from Gaussian probability function, *m_d_* is particle mass, and Δt is the time-step. The diffusion coefficients are calculated using the Stokes–Einstein equation
(3)$${\tilde{D}}_{p}=\frac{k_{B}TC_{c}}{3\pi \mu d_{p}}$$
where *k_B_* is the Boltzmann constant (=$1.38\times 10^{-16}$ cm^2^g/s) and T is the alveolar temperature.

### 2.3. Numerical Methods

ANSYS Fluent (Canonsburg, PA, USA) with the discrete phase model and dynamic mesh was implemented to compute the tidal airflow and particle dynamics. In-house C and Fortran codes were developed to generate injection particle files, define alveolar wall motions, calculate Brownian motion force, and quantify spatial and temporal deposition fractions [54,55]. ANSYS ICEM CFD (Ansys, Inc., Canonsburg, PA, USA) was used to generate the mesh. One-way coupling from the airflow to the nanoparticles was assumed. A grid independent study was performed by testing different mesh sizes. Mesh-independent results were assumed when the discrepancy in the total deposition rate was smaller than 1%. The final alveolar model contained 1.2 million cells. Minitab analysis software (State College, PA, USA) was used to investigate variability in deposition results.

## 3. Results

### 3.1. Airflow Field

The airflow velocity field is shown in Figure 2 at the peak inhalation. Due to the laminar flow nature, the velocity fields resemble each other among different instants in either inhalation cycle or exhalation cycle. The left panel shows the velocity contour of the middle plane. The maximum speed in the alveolar model is around 1 mm/s, in comparison to the boundary-motion speed of approximately 0.03 mm/s. The airflow speed in the first alveolus is in the order of 0.3 mm/s, which is an order of magnitude higher than that in the three neighboring alveoli. The middle panel shows the stream traces of the airflow at the peak inhalation. The velocity magnitude changes a lot along each stream trace. The vena contracta effect occurs in the vicinity of the pores, where the airflow velocity increases and then decreases dramatically before and after the pore apertures. Collateral ventilation is observed from the bottom alveolus (Sac 4) to the two lateral alveoli (Sac 2 and Sac 3), indicating a transient pressure imbalance among the alveoli. Furthermore, plotting stream traces in a time series reveals that the airflow is generally reversible during inhalation and exhalation, with the inspiratory streamline retracing back during exhalation (figure not shown). The instantaneous coherent structures at the peak inhalation are shown in the right panel of Figure 2 in terms of the lambda-2 criterion at the iso-surface of 2.0. A coherent structure is a large-scale defined structure of particular shape that conserves its spatial and temporal features for a long time relative to the vorticity timescale. Coherent structures control the mass, momentum, and heat transfer at large scales and are responsible for flow stability and mixing in local regions. At the peak inhalation, the coherent structures are concentrated in the first alveolus and downstream of the pores. There are also sporadic structures in the duct. This is consistent with the observation that the velocity gradients are highest in these regions. Inhaled particles in these regions will experience quick speed variations, and therefore the transient effect will be most noticeable for large nanoparticles, whose inertia becomes more important.

To study the pore size effect, the velocity contour and stream traces in three different models (pore size of 20 µm, 80 µm, and no septum) were compared during inhalation (Figure 3a) and exhalation (Figure 3b). As noted in Figure 2, the flows in the three alveolar models are generally reversible within one breathing cycle, as evidenced by the similar stream-trace patterns between inhalation and exhalation. In light of the pore size effect, a stronger vena contracta effect was observed in the 20-µm-pore model (left panel, Figure 3), while the jet-like flow is absent in the 80-µm-pore model (middle panel, Figure 3) and the model with no septum (right panel, Figure 3). Interestingly, the collateral ventilation in the 20-µm-pore model, like the 40-µm-pore model, has the inhaled air flowing from the bottom alveolus to the two lateral alveoli, while in the 80-µm-pore model, the inhaled air flows from the two lateral alveoli to the bottom alveolus (Figure 3a vs. Figure 3b). In the alveolar model with no septal wall, the speed of the inhaled airflow constantly decreases from the inlet to the moving wall due to volume expansion. The streamlines are also smoother (less curved) than those in models with varying pore sizes.

### 3.2. Particle Motion

The particle dynamics in the 40-µm-pore model is shown in Figure 4 for 10 nm and 200 nm particles by tracking five particles released from the middle of the duct. The trajectories of the particles are controlled by both wall-expansion-induced convention, particle gravity, and Brownian motions. The random motions are apparent for both particle sizes considered. The amplitudes of the random motion are larger for 10-nm particles than 200 nm particles, as evidenced by the larger speed range for 10 nm particles. It is noted that the instantaneous particle speeds can be two to three orders of magnitude higher than the wall expanding speed, a zero to three orders of magnitude higher than the airflow speed. The instantaneous particle speed fluctuates a lot. Once it touches the wall, it will deposit there, as illustrated by the red arrows in Figure 4a. Because the diffusivity of a 10-nm particle is much higher than that of a 200-nm particle, earlier and higher deposition is expected for 10-nm particles than 200-nm particles.

Figure 5 shows a time series of snapshots of particle positions in the 40-µm-pore alveolar model during the first respiration cycle. Particles were 200 nm in size and were released 0.2 s from the second breathing cycle (T0 = 3.20 s). The first cycle (0–3 s) was simulated to establish the transient airflow field. Due to particle Brownian motion and convection, some particles exited the geometry during exhalation. The Peclet number of the particles are within the range of 0.01–2618. In comparison to the well-defined particle front observed for micrometer aerosols [34], the nanoparticles appear more random and dispersed in distribution. After 0.2 s of release (T1 = 3.40 s), the particles enter the first alveolus and exhibits random velocities on top of the convective main inspiratory flow (Figure 5a). At T2 = 3.80 s (Figure 5b), particles start entering the three neighboring alveoli. Due to the center positions of the inter-septal pores in the septal walls, the side view of the particles exhibits a triangle shape (Figure 5b, right panel), indicating a stronger transport effect from the bulk convective flow. The vena contracta effect of the pores accelerates the flow. Once the particles passed the pores, the velocity inside the alveoli decreases and diffusion becomes dominant (Figure 5c). During the exhalation phase (Figure 5e,f), most particles remain inside the alveoli. Only a small fraction of particles retrace back and exit the alveoli. Considering that the tidal volume is approximately 23% of the functional residual capacity (FRC), only 18.7% (0.23/1.23) of the air in the alveoli at the end of inhalation was exhaled. While the particles are well mixed due to the molecular diffusion, only a small portion (close to 18.7%) was exhaled and the majority of inhaled particles remain in the alveoli. This very irreversibility of the nanoparticles (not airflow) is the primary mechanism responsible for their deposition in the acinar region. While trapped within the alveoli, most of these particles will deposit after several cycles through a combined effect of diffusion and sedimentation.

To investigate the effect of dynamic wall motion on nanoparticle deposition, particle behaviors were compared between alveolar models with and without wall motion (Figure 6). Different from the micrometer particles that precipitate due to gravity in the absence of wall motion [34], nanoparticles remain suspending in the proximity of their release positions and continuously deposit onto the duct wall (Figure 6a). Figure 6b shows the deposition rate in the duct as a function of time. After one second, most particles deposit and nearly all particle deposition after 1.8 s (T = 5.20 s). Moreover, most of the particles deposit in the lower half of the duct due to gravity, while a small fraction of particles deposit in the upper half of the duct or exit the geometry (Figure 6c).

To evaluate the effect of the pore size on particle dynamics, snapshots of particle positions at three instants in the 80-µm-pore alveolar model were plotted in Figure 7. In comparison to those in the 40-µm-pore alveolar model as shown in Figure 5, a shorter distance of particle advancement was observed in the 80-µm-pore alveolar model at each corresponding instant, due to the reduced vena contracta effects. At T2 = 3.80 s (Figure 7a), most of the particles are still in the first alveolus of the 80-µm-pore alveolar model. Instead, an appreciable portion of particles enters the three contiguous alveoli of the 40-µm-pore alveolar model (Figure 5b). At T3 = 4.10 s, particles are less dispersed (more concentrated) in the 80-µm-pore alveolar model (Figure 7b) than those in the 40-µm-pore alveolar model (Figure 5c). Likewise, a less dispersed pattern was observed in the 80-µm-pore alveolar model during exhalation (T5 = 5.20 s, Figure 7c vs. Figure 5e). In particular, few particles enter the fourth alveolus (Sac 4) due to the smaller pore size that communicates the first and fourth alveoli. These differences indicate that one of the major influences from the pore size is the vena-contracta-related convection, as well as in determining the resistance and pressure distributions among the connecting alveoli.

### 3.3. Particle Deposition

Figure 8 shows the deposition distribution of inhaled nanoparticles in alveolar models with varying pore sizes (20 µm, 40 µm, and 80 µm) for 10 nm and 200 nm aerosols. Surprisingly, the deposition patterns for the six cases considered herein look very similar to each other, indicating a predominating Brownian motion effect for particles less than 200 nm. However, the deposition patterns are also highly heterogeneous, indicating a notable sensitiveness of local deposition to the alveolar shape and size. For all cases presented in Figure 8, an elevated deposition rate was noted in the duct, a phenomenon that is similar to the “diffusional screening” of inhaled gas [56], where significant O_2_ absorption occurs in the acinus entrance and proximal alveoli, causing a constant decrease in O_2_ concentration along a serially arranged alveoli. There are three characteristic length scales in the alveolar models considered in this study: The duct, the alveoli, and the pore. It is expected that significant diffusional screening occurs in the duct and around the pores. As the pore size increase, the diffusional screening effect diminishes, which promotes a less heterogeneous (more uniform) distribution of particle deposition (Figure 8a vs. Figure 8c).

The temporal variation of the deposition fractions in different parts of the alveolar model are shown in Figure 9, which includes the duct, the four alveoli (Sac 1–4), and the necks that connect the alveoli. The high deposition rate in the duct is reflective of the importance of the diffusional screening, where the random motion of nanoparticles promotes significant deposition. This effect is most pronounced for ultrafine particles (10 and 50 nm in Figure 9a,b) that have higher diffusivities and diminishes with increasing particle sizes (200–800 nm in Figure 9c–f). Similar deposition fractions were observed between the two lateral alveoli (Sac 2 and Sac 3) for all cases considered because of their geometrical symmetry relative to the Sac 1, even though the deposition fraction varied with the particle size.

An abrupt DF increase in Sac 2 and Sac 3 occurs for 800 nm particles (Figure 9f vs. Figure 9a–e), presumably due to an increasing gravitational effect. The DF in Sac 4 constantly increases with the particle size, which becomes equivalent in magnitude with the two lateral alveoli since 500 nm. The other particle-size-dependent difference is the time the particles take to complete the deposition, which increases with particle size (Figure 9). For instance, it took slightly one cycle for all 10-nm particles to deposit, while took approximately four cycles for 800-nm particles.

The surface deposition of large nanoparticles in the 40-µm-pore model is further visualized in Figure 10 for 500 nm and 800 nm aerosols. Compared to 10 nm and 200 nm aerosols in Figure 8b, particles of 500 nm and 800 nm are more concentrated at the bottom of each alveolus and close to the axial centerline, while at the same time, fewer particles deposit in the duct. This observation is even more pronounced for 800 nm aerosols than 500 nm aerosols. Moreover, a significantly higher fraction of 800 nm particles deposit in the bottom alveolus (Sac 4) than 500 nm particles due to increased gravitational effects, as also illustrated in Figure 9f vs. Figure 9d.

The final deposition fraction in the 40-nm-pore alveolar model as a function of particle sizes ranging from 1 nm to 1000 nm is shown in Figure 11a. The sub-regional deposition fractions (DF) in the six sections (duct, Sac 1–4, and neck) were plotted by stacking their DF values along the serial direction, the sum of which gave the total DF in the alveolar model. For comparison purposes, the corresponding plot in the 80-µm-pore alveolar model is presented in Figure 11b. It is noted that the DF is based on a bolus aerosol released into the geometry 0.2 s after the start of the second cycle, not a continuous inhalation.

Several observations are noteworthy in Figure 11. First, DF in the cylinder (or duct, gray bar) is dominant for particles smaller than 100 nm due to the diffusional screening effect, and constantly decreases for larger submicron particles. Second, the DF in Sac 1 (green bar) appears to be relatively independent of both particle size and pore size and has a magnitude of approximately 20%. This independence was also observed in the two lateral alveoli (Sacs 2 and 3, blue and red bars), but only for particles with diameters smaller than 200 nm. For larger particles, DF in these two sacs slowly increase with the particle size. A similar trend was also noted in Sac 4, but in a much dramatic way, which is reasonable because Sac 4 is directly beneath Sac 1 and therefore, will bear more influence from the gravitational sedimentation. Third, the impact from the pore size mostly manifests itself in sac4 and the necks (Figure 11a vs. Figure 11b). The lower DF in 80-µm-pore model for ultrafine particles (1–100 nm) is due to the lack of vena contracta effect through the pore connecting Sacs 1 and 4, while the higher DF for large nanoparticles (600–1000 nm) comes from a larger area in the 80-µm-pore model for effective gravitational sedimentation. The higher DF in the necks are simply from the larger surface area of the necks in the 80-µm-pore model than that in the 40-µm-pore model (Figure 11b vs. Figure 11a, the top bar).

To further investigate the impact of the pore size on nanoparticle deposition, DFs in models with different pore sizes (20 µm to no septum) were plotted in Figure 12a,b for 10 nm and 200 nm, respectively. Considering the 200-nm aerosols, which is typical for inhalation nanomedicines, the pore size appears to exert an insignificant impact on both the total and sub-regional deposition fractions. A box plot that shows the DF variability caused by the pore size is also given on the right panel of Figure 12a,b. As noted before, the DF progressively decreases from the proximal to distal (i.e., from cylinder to Sac 4) no matter what the pore size is, corroborating the diffusional screening effects for both 10 nm and 200 nm aerosols. Among the six sub-regions considered, the variability is highest in the cylinder (duct) and Sac 4. The deposition in the duct is dominated by diffusion, while in the Sac 4 is more dictated by the gravitational sedimentation.

## 4. Discussion and Summary

### 4.1. Effects of Wall Motion on Nanoparticle Deposition

It is well accepted that the deposition mechanics of nanoparticles in the acinar region are diffusion and gravitational sedimentation. The contributions from the wall-motion-induced convection, however, are not clear. Will it be indispensable to nanoparticle deposition, and how important it is relative to the other two mechanisms? Even though this study was not devoted to this question, some conclusions can be made from the results in this and other studies [34,41,42]. From Figure 6, the inhalation flow induced by the alveolar wall expansion is necessary to carry the particle-entrained bulk flow into the alveolar space to deposit there. Without wall motion, nearly all particles will deposit in the alveolar duct due to the diffusional screening effect. It is thus inferred that the timescale for particles to pass through the alveolar duct will directly associate with the deposition rate in the duct, as well as the fraction that enters the alveoli. However, further studies are needed to examine the influences of the wall motion kinematics on the level of aerosol irreversibility and quantify the relationship between the wall motion and alveolar deposition. This observation also calls into question the accuracy of using inhaled particles to measure the alveolar dimension in vivo [57].

### 4.2. Effects of Particle Size on Nanoparticle Deposition

The differences in transport and deposition in the alveoli between micron- and nanoparticles are briefly discussed below. For repairable micron particles (1–3 µm) that has negligible diffusivity, particles closely follow the respiratory flows and a well-defined aerosol front is often observed that visualize the carrier airflow [34]. The surface deposition of micron particles is also more localized, which concentrated near the inter-septal pores and the bottom of the alveoli. In contrast, nanoparticles exhibit more dispersed patterns in both transport and deposition. The smaller the particle, the higher the degree of dispersion will be. Nanoparticles are found to deposit anywhere within the alveoli, even though not uniformly. The random motions of nanoparticles greatly enhance their mixing with the residual air and lead to varying degrees of aerosol irreversibility peculiar to the size of the nanoparticles. In this study, we noticed that the respiratory flows are close to symmetric and reversible, with streamlines largely retaining their positions during inhalation and retracing their directions during exhalation. However, aerosol irreversibility was observed in both micron and nanoparticles, which is much more significant for nanoparticles than the micrometer particles. Likewise, the aerosol irreversibility increases with decreasing particle sizes for nanoparticles, which retains more particles in the alveoli during exhalation and give rise to higher deposition rate and more uniform deposition pattern.

Sub-regional deposition rates are found to be sensitive to the size of nanoparticles (Figure 11). In the duct-alveolar model considered in this study, this sensitivity is most pronounced in the alveolar duct (cylinder) and the bottom alveolus (Sac 4), while is much less significant in the two lateral alveoli (Sac 2 and Sac 3) and the necks. The progressive decline in the cylinder deposition with increasing particle size is attributed to the diminishing diffusional screening effect, while the sharp increase of Sac 4 deposition rate from 600 nm to 1000 nm particles results from the gravitational sedimentation, which is proportional to the cubic of the particle diameter. The deposition rate in the lateral alveoli (Sac 2 and Sac 3) is almost independent of the particle size for particles smaller than 500 nm and shows a slight increase with particle size after that.

It is also interesting to note that the particle size with the minimum deposition is different in models with different pore sizes. It is 600 nm in the 40-µm-pore model and 500 nm in the 80-µm-pore model. Even though we are not clear at this moment what factors cause this difference, it can largely be attributed to the interplay between the diffusive particles and the complex alveolar geometry, including the diffusional screening effect in the duct, the vena-contracta jet-like effect in the pore, and the diffusion-driven mixing in the alveolar space.

### 4.3. Effects of Inter-Septal Pore Size on Nanoparticle Deposition

Surprisingly, the impacts from the inter-septal pore size is insignificant on both total and sub-regional deposition of nanoparticles in the duct-alveolar models considered in this study. This similarity may partially result from an identical V_T_/FRV ratio of 0.23 implemented for all models. A smaller pore size may strengthen the vena-contracta jet-like effect and increase the flow impedance through the pore. However, the quick volume expansion before and after the pore may restrict the vena-contracta effect and jet-induced mixing only to the close proximity of the pore. Moreover, the constant V_T_/FRV ratio gives rise to the same magnitude of wall expansion/contraction, which brings in and expels out the same volume of air. Based on a well-mixed aerosol inside the alveoli that is more dictated by the diffusion (particle size) than the convection (pore size), a similar number of particles are expected to be exhaled out regardless of the inter-septal pore sizes. In a recent study, Hofemeier et al. [40] also reported that variation in the acinar geometrical heterogeneity had an insignificant impact on the total acinar deposition.

The inter-septal pore size is closely relevant to alveolar health, which enlarges in the presence of pulmonary diseases, such as asthma and COPD. In severe conditions, the inter-alveolar septum will break down either partially or entirely, as seen in emphysema. Results in this study indicate that the delivery efficiency may not be significantly affected as long as the patients can perform the required breathing maneuvers, in this case, a given tidal volume.

#### Elevated Nanoparticle Deposition When Released at Early Inhalation

One unexpected observation is that, when nanoparticles were released into the acinar region at the start of the inhalation, high deposition rates (74–95%) were predicted for all diffusive particles by excluding the deposition in the upstream airways (i.e., from the mouth to the respiratory bronchioles). This may initially contradict the long-accepted understanding that nanoparticles, especially 500–600 nm particles, do not deposit effectively in the lung and most inhaled dose will be exhaled out of the airway. This traditional understanding was based on two assumptions: (1) Deposition was considered in the whole respiratory tract that includes both the conducting airways and distal alveoli, and (2) nanoparticles were inhaled continuously throughout the entire inhalation cycle. One unique feature in alveolar ventilation is that dead space occupies more than 80% of the acinar region, while there are much smaller dead spaces in the conducting airways. Based on a 23% ratio of the tidal volume over FRV (functional residual volume) under normal breathing, the inhaled air only accounts for 18.7% (0.23/1.23) of the total volume of the alveolus at the end of the inhalation. When nanoparticles enter the alveoli, they mix with the residual air and fill the entire space heterogeneously depending on the nanoparticle diffusivity. Upon expiration, only 18.7% of the air will be exhaled and carries the suspending nanoparticles in it. The concentration of nanoparticles is generally higher near the alveolar entrance, which primarily depends on the nanoparticle diffusivity. However, overall, the majority of nanoparticles that have already mixed with the residual air will remain and eventually deposit either through diffusion or sedimentation.

The second factor is the time point when nanoparticles enter the duct-alveoli geometry. Particles that reach the alveolar duct at the beginning of inhalation have more time to enter the terminal alveoli and penetrate deeper in the alveolar airspace. By contrast, particles that reach the duct at the end of inhalation penetrate much shallower. Moreover, they most likely do not have sufficient time to mix with the residual air and will be exhaled out during wall contraction.

The observation that elevated deposition is possible when nanoparticles are released at the beginning of inhalation may have significant implications in systemic delivery of pulmonary nanomedicines. By optimizing the time to release drugs into the mouth, the maximum pulmonary dose can be achieved, or the control of pulmonary delivery efficiency can be realized. The optimization can also include factors that are related to drug carriers (such as particle size, charge, hygroscopic property), patients (breathing depth, frequency, holding), and devices (nebulizer, metered dose inhaler, dry powder inhaler), to name a few. Enhanced dosing of pulmonary nanomedicines will reduce dosing frequency and improve patient compliance. However, such optimization should be performed in an integrated manner by considering the whole respiratory tract in opposite to the acinar region or conducting airway only.

### 4.4. Limitations

Limitations of this study that may affect the physical realism of the results include idealized alveolar model, simplified wall kinematics, and non-continuous inhalation of particles. Particle charge [58,59], size [60], and hygroscopy effect [61] were also neglected. Microscopy examinations have disclosed an intricate morphology of the in vivo pulmonary alveoli with polyhedral airspaces packed like a fractal [62,63,64,65,66]. The thickness of the interalveolar septal wall also varies [67]. The duct-alveolar model in this study was reconstructed from simple shapes (cylinders and spheres) with constant wall thickness. There was no more than one pore in one septum in this study, while there can be one to seven pores in life conditions [68]. In healthy lungs, the pore size ranges from 2 µm to 15 µm [18,68]. Likewise, the alveolar size can be different. Further studies of the alveolar models with smaller and multiple pores, as well as of different alveolar sizes, are warranted. Second, a sinusoidal waveform and an I:E (inspiration: expiration) ratio of 1:1 were adopted for the wall kinematics. A different waveform and I:E ratio may give different deposition results. Thirdly, nanoparticles were released into the alveolar geometry at the start of the inhalation and results hereof do not apply for continuous drug administration. Based on the dilute concentration of inhaled aerosols, particle interactions, such as collision and agglomeration, were not considered. It is also noted that nanoparticles were assumed to deposit on the alveolar walls upon their initial contact. In addition, without considering the respiratory tract above the alveoli, it is still premature to identify the optimal nanoparticle size for pulmonary delivery of nanomedicines. Future studies are needed that include the respiratory tract from the mouth to the terminal alveolar sacs, as well as particle biokinetics after their deposition.

In summary, the behavior and fate of inhaled nanoparticles were numerically investigated in the duct-alveolar model with different pore sizes and rhythmic wall motions. Specific findings are:Rhythmic alveolar wall motion is indispensable for inhaled nanomedicine to deposit in the distal alveoli.Diffusion-driven mixing of inhaled nanoparticles with residual air in the alveoli (termed as aerosol irreversibility) is the primary mechanism that keeps particles from being exhaled out of the alveoli.Concentrated deposition occurs in the alveolar entrance and pore size due to diffusional screening effect.Elevated deposition fractions (~85%, as the ratio of particles depositing in the alveolar model over the particles entering the alveolar model) were predicted for nanoparticles released at the beginning of the inhalation. Improved pulmonary doses of nanomedicine can be achieved by optimizing their release time.The intra-septal pore size was found to have an insignificant impact on nanoparticle deposition on both the total and sub-regional basis, indicating a potentially consistent dose of nanomedicine if the patient can perform the required breathing maneuver.

## Figures and Tables

**Figure 1 nanomaterials-09-01126-f001:**
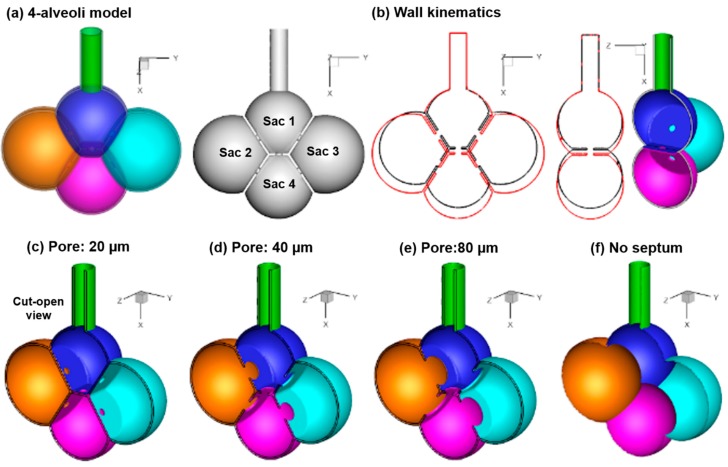
Models: (**a**) Four-alveolar model with septal walls communicated via interalveolar pores, (**b**) alveolar kinematics with rhythmically expanding (red) and contracting (black) walls. Different pore sizes were considered: (**c**) 20 µm, (**d**) 40 µm, (**e**) 100 µm, and (**f**) no septum.

**Figure 2 nanomaterials-09-01126-f002:**
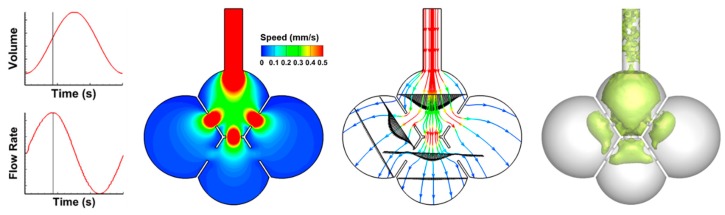
Airflow field in the 40-µm-pore alveolar model at the peak inhalation in terms of velocity contour (**left**), steam traces (**middle**), and coherent vortex structures (**right**).

**Figure 3 nanomaterials-09-01126-f003:**
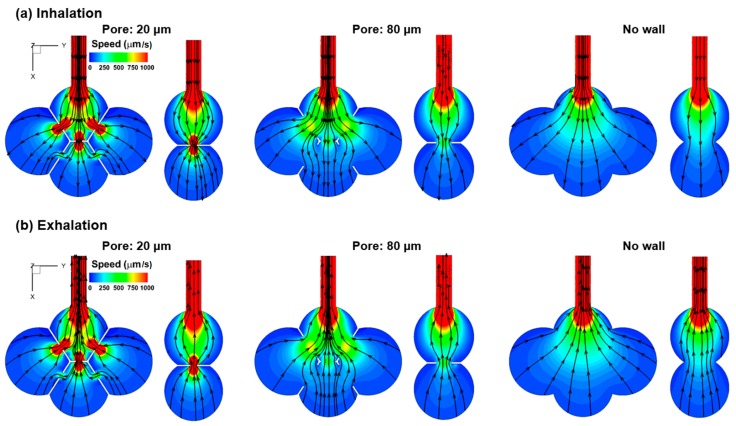
Contour and stream traces in the four-alveolar models with a pore size of 40 µm, 80 µm, and no septal wall during (**a**) inhalation, and (**b**) exhalation.

**Figure 4 nanomaterials-09-01126-f004:**
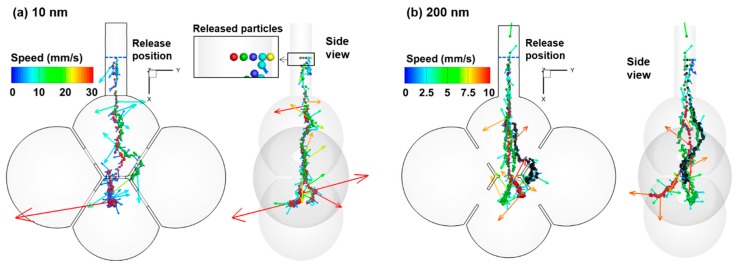
Brownian motion of nanoparticles with a diameter of (**a**) 10 nm and (**b**) 200 nm with their instantaneous velocities superimposed. Five particles were released from the middle of the duct. The pore size of the alveolar model is 40 µm.

**Figure 5 nanomaterials-09-01126-f005:**
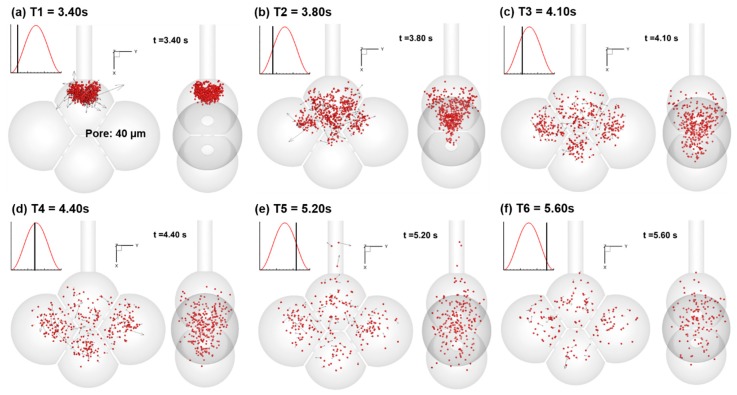
Snapshots of particle positions in the 40-µm-pore alveolar model during the first cycle at different instants: (**a**) T1 = 3.40 s, (**b**) T2 = 3.80 s, (**c**) T3 = 4.10 s, (**d**) T4 = 4.40 s, (**e**) T5 = 5.20 s, and (**f**) T6 = 5.60 s. Particles were 200 nm in diameter and were released 0.2 s (T0 = 3.20 s) from the second breathing cycle. The first cycle (0–3 s) was simulated to establish the transient airflow field. Due to particle Brownian motion and convection, some particles exited the geometry during exhalation.

**Figure 6 nanomaterials-09-01126-f006:**
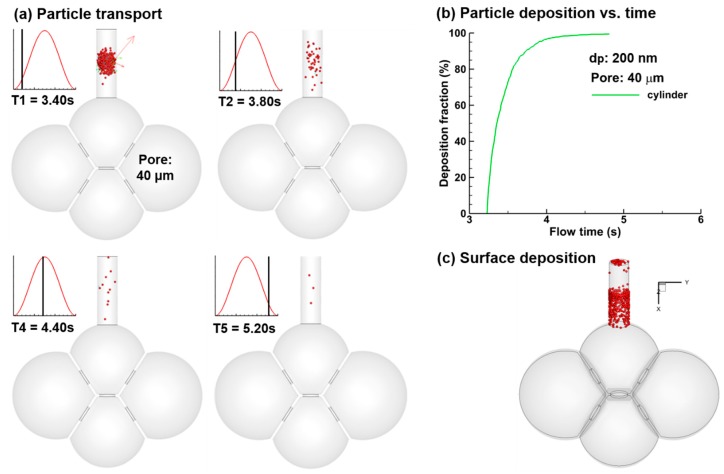
Dynamic when wall kinematics was neglected: (**a**) Particle motions at varying instants, (**b**) time evolution of particle deposition, and (**c**) particle deposition distribution.

**Figure 7 nanomaterials-09-01126-f007:**
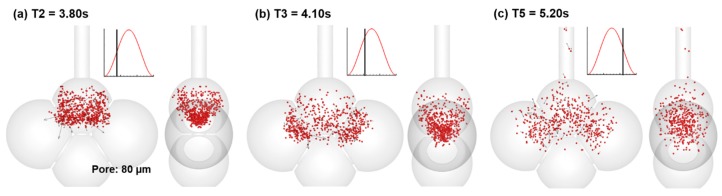
Snapshots of particle positions in the 80-µm-pore alveolar model during the first cycle at different instants: (**a**) T2 = 3.80 s, (**b**) T3 = 4.20 s, and (**c**) T5 = 5.20 s. In comparison to the snapshots of particle positions in Figure 5, a shorter distance of particle advancement was observed in the 80-µm-pore alveolar model at each corresponding instant in the 40-µm-pore alveolar model in Figure 5.

**Figure 8 nanomaterials-09-01126-f008:**
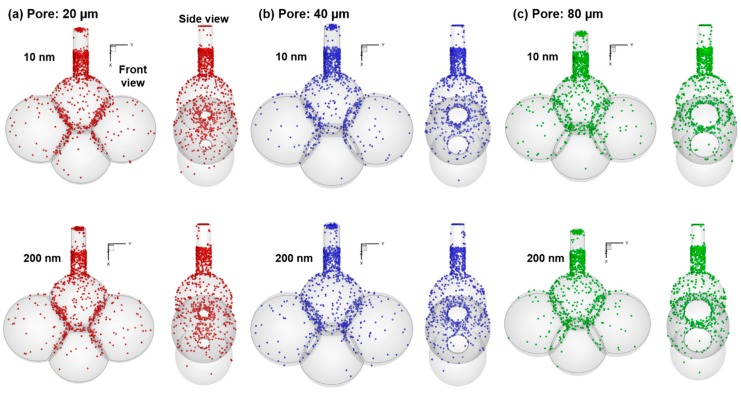
Of surface deposition for 10 nm (upper row) and 200 nm (lower row) particles in the alveolar models with a pore size of (**a**) 20 µm, (**b**) 40 nm, and (**c**) 80 nm.

**Figure 9 nanomaterials-09-01126-f009:**
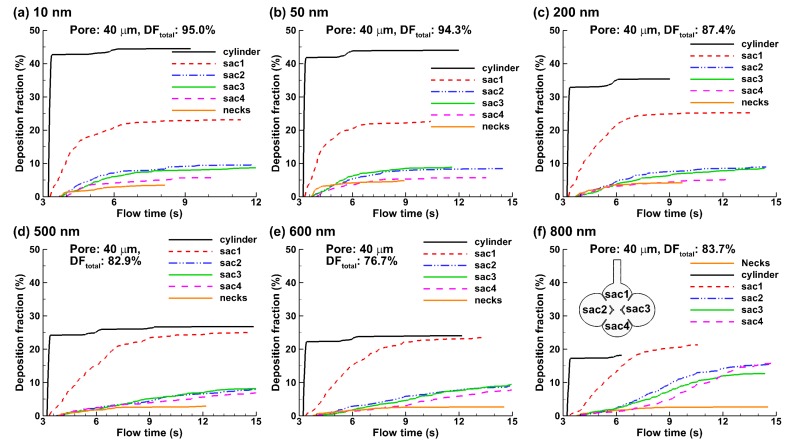
Deposition rate with time in the 40-µm-pore alveolar model for nanoparticles of different diameters: (**a**) 10 nm, (**b**) 50 nm, (**c**) 200 nm, (**d**) 500 nm, (**e**) 600 nm, and (**f**) 800 nm. Both total and sub-regional deposition fraction were shown. Particles were released as a bolus 0.2 s after inhalation (T0 = 3.20 s).

**Figure 10 nanomaterials-09-01126-f010:**
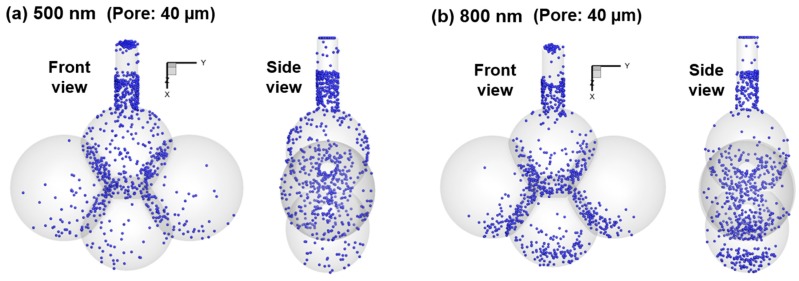
Of surface deposition in the 40-µm-pore model for large nanoparticles: (**a**) 500 nm, and (**b**) 800 nm.

**Figure 11 nanomaterials-09-01126-f011:**
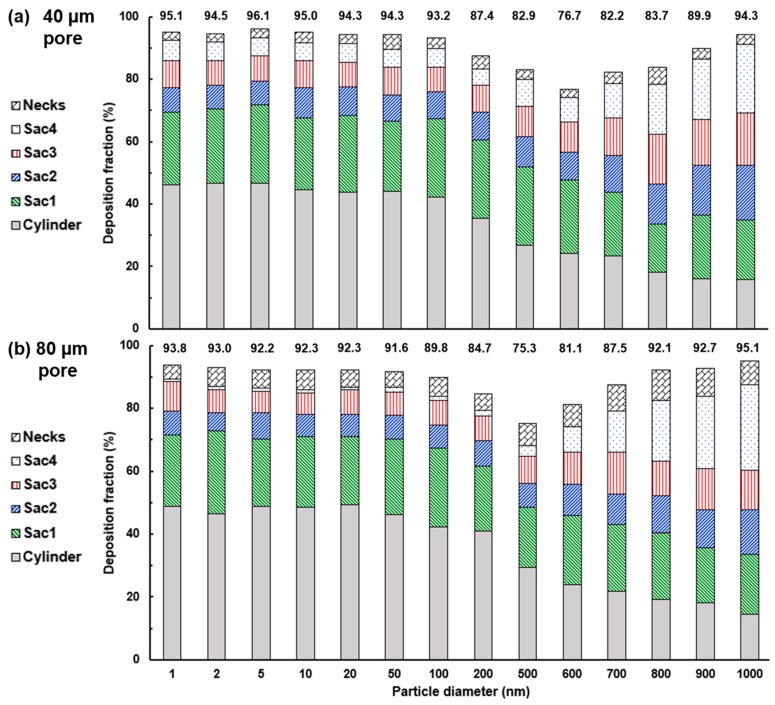
Of total and sub-regional deposition fractions as a function of particle diameter between (**a**) 40-nm-pore and (**b**) 80-nm-pore alveolar models.

**Figure 12 nanomaterials-09-01126-f012:**
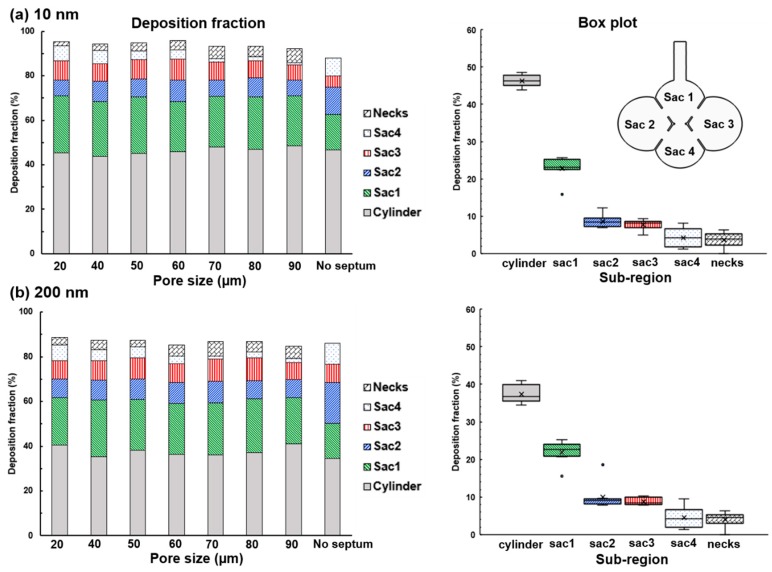
Of particle deposition in the alveolar models with different pore sizes: (**a**) 10 nm, and (**b**) 200 nm. Box plots of the sub-regional deposition fractions were shown in the right panels for 10 nm and 200 nm, respectively.

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
