# Peer review of "Nanoparticle Deposition in Rhythmically Moving Acinar Models with Interalveolar Septal Apertures"

_nanomaterials, 2019, doi:10.3390/nano9081126_

Round 1
Reviewer 1 Report
The manuscript presents the nanoparticle deposition results in a human alveolar duct model using computational fluid dynamics. The modeling approach is sound and the manuscript is well written. The model results are interesting to researchers working in the field of pulmonary diseases and medicines. The following are some minor comments where I would have appreciated additional explanation/clarification.
In the abstract it is mentioned that the alveolar deposition rates are high -" When aerosols were released into the alveoli at the beginning of the inhalation, high alveolar deposition rates (74-95%) were predicted". It might be confusing to the readers as it is not clear why the deposition rates are very high. It would make it clear if the sentence also includes an explanation indicating that the deposition in the upper airways are not considered.
page 7 230 -" In comparison to the well-defined particle front observed for micrometer aerosols,... "- since the micrometer aerosols are not considered in this study, there should be a citation.
line 334 - "the 80-nm-pore alveolar model" - it should be 80 micrometer
line 348 - "manifests itself in sca4 and" - sac4 misspelling
Author Response
In the abstract it is mentioned that the alveolar deposition rates are high -" When aerosols were released into the alveoli at the beginning of the inhalation, high alveolar deposition rates (74-95%) were predicted". It might be confusing to the readers as it is not clear why the deposition rates are very high. It would make it clear if the sentence also includes an explanation indicating that the deposition in the upper airways are not considered.
Response: We are thankful for the Reviewer’s insightful comment on the high alveolar deposition rates. The fact that the exclusion of the deposition in the upper airways led to high deposition rates in the terminal alveolar sacs has now been clarified in Both Abstract and Discussion, as follows:
(Abstract, lines 26-29): “When deposition in the upper airways were not considered, high alveolar deposition rates (74-95%) were predicted for all nanoparticles considered (1-1,000 nm), which were released into the alveoli at the beginning of the inhalation.”
(Discussion, lines 435-437): “One unexpected observation is that, when nanoparticles were released into the acinar region at the start of the inhalation, high deposition rates (74-95%) were predicted for all diffusive particles by excluding the deposition in the upstream airways (i.e., from the mouth to the respiratory bronchioles).”
page 7 230 -" In comparison to the well-defined particle front observed for micrometer aerosols,... "- since the micrometer aerosols are not considered in this study, there should be a citation.
Response: A reference that simulated micrometer aerosols in the alveolar models [34] was added in the text.
line 334 - "the 80-nm-pore alveolar model" - it should be 80 micrometer.
Response: The typo “80-nm-pore” was corrected to “80-µm-pore”.
line 348 - "manifests itself in sca4 and" - sac4 misspelling
Response: The typo was corrected.
Reviewer 2 Report
In recent times, a number of theoretical and numerical studies have investigated airflow and particle deposition in the pulmonary system. However, most earlier studies have excluded the interalveolar septal walls and apertures in their models for geometrical simplicity. In addition, reports of NPs transport and deposition in alveolar sacs with moving walls are infrequent. In this present manuscript the authors aimed to investigate the nanoparticle transport and deposition in a simplified alveolar model, and they summarized their specific findings. This kind of modeling study is required to improve the understanding of deposition of inhaled particles. In particular this manuscript may support the pulmonary nanomedicine translation and encourages the studies related to nanotoxicology. Over all this manuscript is suitable for publication in the journal of nanomaterials as in the present form.
Author Response
Response: We thank the Reviewer for his/her enthusiastic support of the merits of this work.
Reviewer 3 Report
Xi et al. investigated the nanoparticle deposition in acinar models with interalveolar septal apertures. The pore size notably affected the deposition pattern of inhaled nanoparticles but exerted a low impact upon the total deposition fractions. I have some questions regarding this study.
1. Introduction is too long. Authors should describe introduction concisely.
2. Can this soft simulate an uptake to alveolar epithelial cells or uptake to capillary blood vessel?
3. Alveolar size is different in human. Can this soft simulate the results by change the alveolar size?
4. Can this soft simulate the results when one of the alveoli is injured?
5. Authors should show the best nanoparticle in clinical setting.
Author Response
1. Introduction is too long. Authors should describe introduction concisely.
Response: Text in Introduction has been revised wherever appropriate to make it concise. Sentences that have been removed from the original manuscript include: lines 38-43, lines 73-74, and lines 93-94.
2. Can this soft simulate an uptake to alveolar epithelial cells or uptake to capillary blood vessel?
Response: The computational model in this study cannot simulate particle uptake to alveolar epithelial cells or capillary blood. The nanoparticles were assumed to deposit on the alveolar walls upon their initial contact, which would neither get back to the airflow, nor be absorbed into the tissue/blood. This limitation and the need to include the biokinetics of deposited particles were acknowledged in Discussion, lines 483-487: “It is also noted that nanoparticles were assumed to deposit on the alveolar walls upon their initial contact… Future studies are needed that include the respiratory tract from the mouth to the terminal alveolar sacs, as well as particle biokinetics after their deposition.”
3. Alveolar size is different in human. Can this soft simulate the results by change the alveolar size?
Response: The Reviewer raised a very insightful question concerning nanoparticle deposition in the alveoli, which can have different sizes. This work has focused on the effects of the inter-alveolar pore sizes on the alveolar deposition and hasn’t considered the alveolar size effects. However, this computational model can be readily modified to simulate nanoparticle deposition in different sizes of alveoli. The limitations of neglecting the alveolar size and the need for future studies have now been acknowledged in Discussion, lines 476-477: “Likewise, the alveolar size can be different. Further studies of the alveolar models with smaller and multiple pores, as well as of different alveolar sizes, are warranted.”
4. Can this soft simulate the results when one of the alveoli is injured?
Response: The computational model developed in this study can be readily modified to simulate the scenarios when one or more of the alveoli is/are injured. This can be achieved by modifying the alveolar geometry and/or alveolar wall motion kinematics.
5. Authors should show the best nanoparticle in clinical setting.
Response: In this work, the respiratory tract above the alveolar sac was not considered, which can filter a significant fraction of inhaled nanoparticles. Without the integration of the upper respiratory tract, we deem that it is premature to recommend the best nanoparticles in clinical settings. This limitation has been acknowledged in Discussion as follows, (lines 484-486): “In addition, without considering the respiratory tract above the alveoli, it is still premature to identify the optimal nanoparticle size for pulmonary delivery of nanomedicines. Future studies are needed that include the respiratory tract from the mouth to the terminal alveolar sacs …”
Round 2
Reviewer 3 Report
I am satisfied with authors' answers.